# Antibody Responses after SARS-CoV-2 Vaccination in Patients with Liver Diseases

**DOI:** 10.3390/v14020207

**Published:** 2022-01-21

**Authors:** Athanasios-Dimitrios Bakasis, Kleopatra Bitzogli, Dimitrios Mouziouras, Abraham Pouliakis, Maria Roumpoutsou, Andreas V. Goules, Theodoros Androutsakos

**Affiliations:** 1Department of Pathophysiology, Medical School, National and Kapodistrian University of Athens, 11527 Athens, Greece; th.bacasis@gmail.com (A.-D.B.); kbitzogli@gmail.com (K.B.); dmouziouras@med.uoa.gr (D.M.); maroump@gmail.com (M.R.); 2Second Department of Pathology, Medical School, National and Kapodistrian University of Athens, 11527 Athens, Greece; apouliak@med.uoa.gr

**Keywords:** SARS-CoV-2, vaccination, liver diseases, cirrhosis, immunosuppression

## Abstract

The novel mRNA-based vaccines against SARS-CoV-2 display encouraging safety and efficacy profiles. However, there is a paucity of data regarding their immunogenicity and safety in patients with liver diseases (PWLD), especially in those with cirrhosis. We prospectively assessed anti-SARS-CoV-2 S-spike IgG antibodies and neutralizing activity in fully vaccinated PWLD (*n* = 87) and controls (*n* = 40). Seroconversion rates were 97.4% (37/38) in cirrhotic PWLD, 87.8% (43/49) in non-cirrhotic PWLD and 100% (40/40) in controls. Adequate neutralizing activity was detected in 92.1% (35/38), 87.8% (43/49) and 100% (40/40) of cirrhotics, non-cirrhotics and controls, respectively. On multivariable analysis, immunosuppressive treatment was negatively correlated with anti-SARS-CoV-2 antibody titers (coefficient (SE): −2.716 (0.634), *p* < 0.001) and neutralizing activity (coefficient (SE): −24.379 (4.582), *p* < 0.001), while age was negatively correlated only with neutralizing activity (coefficient (SE): −0.31(0.14), *p* = 0.028). A total of 52 responder PWLD were reassessed approximately 3 months post-vaccination and no differences were detected in humoral responses between cirrhotic and non-cirrhotic PWLD. No significant side effects were noted post vaccination, while no symptomatic breakthrough infections were reported during a 6-month follow up. Overall, our study shows that m-RNA-based SARS-CoV-2 vaccines are safe and efficacious in PWLD. However, PWLD under immunosuppressive treatment and those of advanced age should probably be more closely monitored after vaccination.

## 1. Introduction

Various studies have investigated the severity of coronavirus disease 2019 (COVID-19) in patients with liver diseases (PWLD) since the beginning of the pandemic [1,2,3,4]. PWLD and particularly those with cirrhosis seem to be in higher risk for severe COVID-19 course and death [5,6,7], making vaccination against the severe acute respiratory syndrome coronavirus 2 (SARS-CoV-2) of the utmost importance [8,9].

The novel mRNA-based vaccines against SARS-CoV-2 display encouraging safety and efficacy profiles [10,11]. Nevertheless, concerns have been raised regarding safety of these vaccines in PWLD and particularly those with cirrhosis, since only a few of them were included in phase I–III vaccine clinical trials. Moreover, patients with cirrhosis display innate and adaptive immune dysregulation, setting the question of hypo-responsiveness to SARS-CoV-2 mRNA vaccines [12]. Available data on antibody responses to these novel vaccines in PWLD are scarce, with most data coming from studies focusing on liver transplant recipients [13,14]. Herein, we prospectively assessed antibody responses in PWLD with and without cirrhosis vaccinated with two doses of mRNA SARS-CoV-2 vaccine.

## 2. Materials and Methods

### 2.1. Study Design

Consecutive patients visiting the hepatology outpatient clinic at “Laikon” General Hospital, Athens, Greece, from 1 March to 31 May 2021, were recruited for assessment of serum anti-SARS-CoV-2 antibodies at 3 time points: before the first vaccine dose, 1 month after the second vaccine dose and 3 months after the second vaccine dose. Inclusion criteria included the presence of chronic liver disease and planned vaccination with the Pfizer-BioNTech BNT162b2 or the Moderna mRNA-1273 vaccines, while exclusion criteria included age < 18 years and prior COVID-19 clinical infection or positive pre-vaccination anti-SARS-CoV-2 antibodies. Demographics, clinical characteristics (etiology of liver disease, presence of liver cirrhosis and comorbidities) and concomitant medications were acquired from patients’ medical records and interviews. A patient was deemed to be cirrhotic when compatible liver biopsy, transient elastography and/or clinical and biochemical profile were present. Vaccination-related details (type of vaccine, vaccination dates and side effects) were recorded retrospectively during patients’ first follow-up visit, 1 month after second vaccine dose, using a structured questionnaire. Recruited PWLD were followed up for a total period of 6 months after the second vaccine dose and long-term side effects or breakthrough SARS-CoV-2 infections were also recorded. Regarding symptomatic breakthrough infections, these were noted at each patient’s follow up visit, up to a 6-month period after the 2nd vaccine dose. For patients that missed their follow up visit, a telephone interview was performed. A cohort of age- and gender-matched individuals without history of chronic liver disease, including friend and/or family members of the patients and healthcare personnel of the affiliated department, was used as a control group.

### 2.2. Measurement of Anti-SARS-CoV-2 Antibodies

Anti-SARS-CoV-2 IgG antibodies against the S1 domain of spike protein of SARS-CoV-2, isolate Wuhan-Hu-1, were detected using an FDA-approved ELISA method (Euroimmun, Lübeck, Germany). The cut-off positive threshold was >1.1 according to manufacturer’s instructions, after calculating the ratio of optical density (OD) of samples measured at 450 nm, divided by the OD value provided by the calibrator. The aforementioned ratio corresponds to anti-SARS-CoV-2 antibodies titers throughout the current manuscript. For the assessment of neutralizing activity of the anti-SARS-CoV-2 antibodies, a cPass ELISA SARS-CoV-2 Surrogate Virus Neutralization Test Kit (GenScript Biotech B.V, Piscataway, NJ, USA), testing antibody-mediated inhibition of wild type SARS-CoV-2 RBD binding to the human host receptor angiotensin-converting enzyme type 2 [15], was used, and a cut-off of >30% inhibitory concentration was considered as positive.

### 2.3. Statistical Analysis

Statistical analysis was performed by SAS software platform (SAS Institute Inc., Cary, NC, USA). Descriptives were expressed as median and range (minimum–maximum), or frequencies and percentages. Analysis of categorical data was performed using the chi-square test with Yates’ correction or Fischer’s exact test when cell counts were <5. For continuous variables, normality was tested with the Shapiro–Wilk test and subsequently either Mann–Whitney U test (Wilcoxon rank sum test) or *t*-test were applied where appropriate. The statistical significance level threshold was set at *p*-value ≤ 0.05 and all tests were 2-sided. In order to identify independent variables that were associated with hampered anti-SARS-CoV-2 antibody levels and neutralizing activity among PWLD and controls one month after the second vaccine dose, multivariable linear regression models were applied.

## 3. Results

### 3.1. Characteristics of PWLD and Controls

A total of 91 PWLD were initially screened for inclusion in the study. Because of positive anti-SARS-CoV-2 antibody testing prior to vaccination in 4 PWLD (4.04%), 87 PWLD, and 40—age- and gender-matched—controls were included in the final analysis. The median age of PWLD was 67 years old (range: 27–86); 44 of them (50.6%) were females. The most common causes of liver disease were chronic hepatitis B (CHB) infection (30, 34.5%), followed by non-alcoholic fatty liver disease (16, 18.4%) and autoimmune hepatitis (14, 16.1%). A total of 38 (43.7%) PWLD were cirrhotic, with the median Child–Turcotte–Pugh (CTP) and model for end-stage liver disease (MELD) scores being 6 (range: 5–11) (25 Class A, 12 Class B and 1 Class C) and 9 (range: 6–25), respectively. A total of 30 PWLD (34.5%) were under immunosuppressive treatment. Baseline characteristics of the cirrhotic PWLD, non-cirrhotic PWLD and controls are detailed in Table 1. The median time of blood collection after the second vaccine dose was 24 (14–57), 25 (15–45) and 24 (16–38) days for cirrhotic PWLD, non-cirrhotic PWLD and controls, respectively (*p* = 0.595).

### 3.2. Antibody Responses One Month Post Vaccination in PWLD and Controls

Proportions of patients with anti-SARS-CoV-2 antibodies above cut-off among cirrhotic, non-cirrhotic PWLD and controls were 97.4% (*n* = 37), 87.8% (*n* = 43) and 100% (*n* = 40), respectively (*p* = 0.027). Similarly, neutralizing activity above cut-off was detected in 92.1% (*n* = 35), in 87.8% (*n* = 43) and 100% (*n* = 40) of PWLD with and without cirrhosis and controls, respectively (*p* = 0.079). The median (range) anti-SARS-CoV-2 antibody titers were 6.26 (0.74–12.13) in PWLD with cirrhosis, 8.02 (0.08–12.52) in PWLD without cirrhosis and 7.65 (3.48–11.58) in controls (*p* = 0.197); meanwhile, the median neutralizing inhibitory concentration in cirrhotic, PWLD without cirrhosis and controls was 89.91% (13.08–99.2), 94.13 (5.87–99.53) and 93.80% (53.6–99.2), respectively (*p* = 0.410) (Figure 1). When excluding PLWD under immunosuppressive treatment, no significant differences were found in seroconversion rates, antibody titers and neutralizing activity levels among all subgroups (Appendix A).

In multivariable analysis, assessing age, gender, immunosuppressive treatment, presence of liver disease and presence of cirrhosis, immunosuppressive treatment was negatively correlated with anti-SARS-CoV-2 antibody titers and neutralizing activity (*p* < 0.001, coefficient (SE): −2.716 (0.634) and *p* < 0.001, coefficient (SE): −24.379 (4.582), respectively), while age was negatively correlated only with neutralizing activity (*p* = 0.028, coefficient (SE): −0.31 (0.14)). Presence of liver disease and/or cirrhosis were not correlated with either lower anti-SARS-CoV-2 antibody titers or neutralizing activity.

### 3.3. Antibody Kinetics Post Vaccination in Responder PWLD

A total of 52 responder PWLD were re-evaluated in a median (range) time duration of 105 (75–130) days after the 2nd dose without any clinical or epidemiologic findings suggestive of SARS-CoV-2 breakthrough infection. Additionally, 5 PWLD out of the 52 (9.6%) had antibodies to SARS-CoV-2 below the cut-off threshold, with all of them having liver cirrhosis. Although a statistically significant decrease in antibody titers was noted over time in both groups (Figure 2), no statistical differences were detected in seropositivity rates of anti-SARS-CoV-2 antibodies (*p* = 0.051) and neutralizing activity (*p* = 0.61) among cirrhotic and non-cirrhotic PWLD. No significant differences were found in anti-SARS-CoV-2 antibody and neutralizing activity levels (*p* = 0.38 and *p* = 0.31, respectively) as well. Similarly, no significant differences were noticed when PWLD treated with immunosuppressants were compared to those who were not.

### 3.4. Safety of the mRNA SARS-CoV-2 Vaccines in PWLD and Controls

Overall, both mRNA-based vaccines had a very good safety profile. Even though 26 of 127 participants (20.5%) reported systemic side effects, all of them were transient and self-limited. More specifically, the most common post-vaccination side effects in our cohort (PWLD and controls) were pain at the injection site (40.16%), fatigue (14.17%), low-grade fever (9.45%) and headache (8%), with no statistically significant difference between PWLD and controls (*p* = 0.33, 0.92, 0.51 and >0.99, respectively). No cirrhotic patients showed post-vaccination liver-related adverse effects, such as acute on chronic liver failure, worsening of ascites or deterioration of liver synthetic capability.

### 3.5. Symptomatic SARS-CoV-2 Infections in Vaccinated PWLD

During a 6-month follow up period after the 2nd dose, no symptomatic breakthrough infections were reported.

## 4. Discussion

Even though PWLD are at a high risk of morbidity and mortality from COVID-19 only a few studies exist regarding the efficacy of SARS-CoV-2 vaccination in these patients. PWLD and particularly those with cirrhosis [12] display immune dysfunction that predisposes patients to bacterial infections, hemodynamic derangement, organ inflammatory damage and poor response to vaccinations [16,17,18,19,20]. Various mechanisms have been proposed for the vaccine related hyporesponsiveness in these patients. More specifically, cirrhosis is characterized by decreased numbers of T helper and cytotoxic T cells, as well as loss of CD27+ memory B cells [20]. As a result, low T helper cell counts could hamper B cell differentiation into plasma cells and proliferation of memory T cells needed for long lasting immunity [21]. Moreover, inadequate generation and circulation of SARS-CoV-2-specific cytotoxic T cells could possibly alter the early viral control [22], while loss of SARS-CoV-2 spike protein-specific memory B cells would impede rapid production of spike-specific antibodies upon a secondary antigen encounter [23]. Therefore, we conducted a prospective, single-center study, trying to address the question of safety and immunogenicity of the novel mRNA-based SARS-CoV-2 vaccines in a cohort of PWLD. 

In our cohort, novel mRNA-based SARS-CoV-2 vaccines were immunogenic in the vast majority of PWLD, consistent with data reported from vaccine trials [10,11]. Moreover, the presence of cirrhosis or chronic liver disease were not associated with mRNA vaccine hypo-responsiveness. Our results are in line with the study by Thuluvath and colleagues [24] among vaccinated PWLD, including cirrhotic patients and liver transplant recipients. In this study only a few PWLD (~4%) did not respond to SARS-CoV-2 vaccination, with no statistically significant differences between cirrhotic and non-cirrhotic PWLD. 

When assessing antibody levels in our cohort, immunosuppressive treatment was negatively correlated with both anti-SARS-CoV-2 antibody levels and neutralizing activity, while age was negatively correlated with neutralizing activity only. On the contrary, the presence of liver disease or liver cirrhosis showed no correlation with antibody titers or neutralizing activity. Regarding age, our findings are consistent with previously published data on the effect of age on both immunogenicity and subsequent sustainability of humoral responses [25,26,27,28]. This could be partially explained by immunosenescence along with enriched comorbidities in older individuals [29]. As far as immunosuppressive treatment is concerned, a variety of studies, mainly on solid-organ transplanted patients and patients with autoimmune rheumatic diseases, have shown the correlation between immunosuppression and antibody formation after SARS-CoV-2 vaccination [13,30,31,32,33,34]. 

Fifty-two PWLD with adequate humoral response one-month after the second vaccine dose were reassessed for titers of anti-SARS-CoV-2 antibodies and neutralizing activity. Even though all patients had a decline in anti-SARS-CoV-2 antibody levels and their neutralizing activity, humoral responses in 90% (the majority) of our PWLD were found to be above the assay’s threshold of positivity. This finding is in line with those found in studies in vaccinated healthcare workers with no liver diseases [35,36]. 

No serious topical or systemic side-effects were noted post vaccination in our cohort, like in most studies addressing this issue [10,11,37,38]. Notably, these side effects were comparable between PWLD, with or without cirrhosis, and controls. Moreover, upon follow up, no worsening of PWLD’s clinical status was noted, making mRNA vaccines a safe option for all PWLD with or without cirrhosis. 

Our study is the first, to our knowledge, assessing both responsiveness as well as sustainability of anti-SARS-CoV-2 antibodies after mRNA-vaccination in PWLD, however, it was not free of limitations. Firstly, the sample size was relatively small, and as a result the proportion of patients with advanced liver cirrhosis (CTP C) are too few to reach robust results in this subgroup of patients; thus, larger studies, with many such patients, are needed. Secondly, although the ELISA-based SARS-CoV-2 surrogate virus neutralization test we used, positively correlates with the other “gold standard” neutralization assays, the correlation has been shown to be modest [39]. As such, the inability to account for synergistic action of antibodies targeting different epitopes, and the limited detection of antibodies that only block the RBD/ACE2 interface and not non-RBD sites, may have led to false-negative results. Thirdly, vaccine-induced cellular immunity against SARS-CoV-2 was not investigated. Fourthly, as new SARS-CoV-2 variants of concern demonstrate mutations particularly in the spike protein, their recognition by neutralizing antibodies from vaccinated individuals may be compromised [40]. Finally, the identification of SARS-CoV-2 post-vaccination breakthrough infections was based on self-reporting.

## 5. Conclusions

The novel mRNA-based SARS-CoV-2 vaccines seem to be effective and safe in patients with liver diseases.

## Figures and Tables

**Figure 1 viruses-14-00207-f001:**
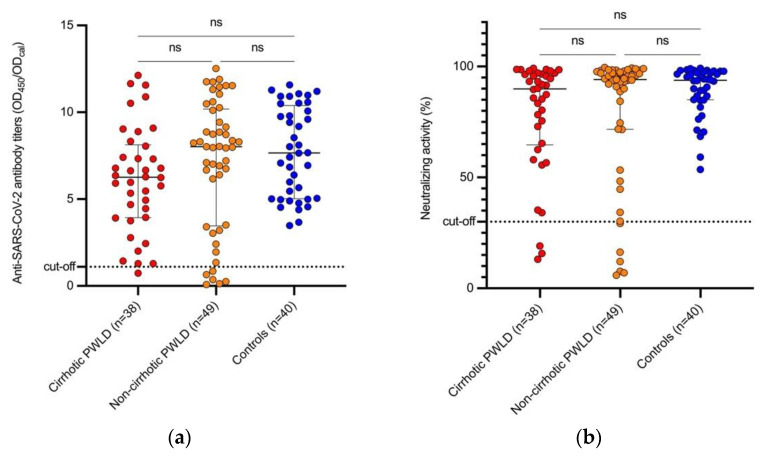
Humoral immune responses one month after the second vaccine dose in cirrhotic PWLD, non-cirrhotic PWLD and controls. (**a**) Anti-SARS-CoV-2 S1-protein IgG antibody titers; (**b**) neutralizing activity. PWLD—patients with liver disease; SARS-CoV-2—severe acute respiratory syndrome coronavirus 2; OD_450_—optical density of serum samples measured at 450 nm; OD_cal_—optical density of calibrator; ns—non-significant.

**Figure 2 viruses-14-00207-f002:**
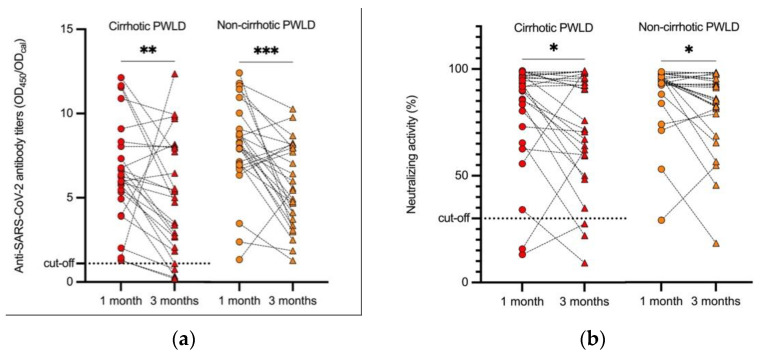
Kinetics of humoral immune responses approximately 3 months after the 2nd vaccine dose in cirrhotic PWLD and non-cirrhotic PWLD. (**a**) Anti-SARS-CoV-2 S1-protein IgG antibody titers; (**b**) neutralizing activity. PWLD—patients with liver disease; SARS-CoV-2—severe acute respiratory syndrome coronavirus 2; OD_450_—optical density of serum samples measured at 450 nm; OD_cal_—optical density of calibrator; ns—non-significant; * *p* < 0.05; ** *p* < 0.01; *** *p* < 0.001

**Table 1 viruses-14-00207-t001:** Demographics, clinical characteristics and vaccination-related details of the study cohort.

	Cirrhotic PWLD (N = 38)	Non-Cirrhotic PWLD (N = 49)	Controls (N = 40)	*p*-Value
Demographic characteristics				
Age (years), median (range)	67 (27–86)	65 (35–81)	71.5 (27–88)	0.300
Female gender, *n* (%)	16(42.1)	28 (57.1)	18 (45)	0.320
Comorbidities				
Diabetes mellitus, *n* (%)	16 (42.1)	11 (22.4)	9 (22.5)	0.080
Pulmonary disease, *n* (%)	4 (10.5)	2 (4.1)	1 (2.5)	0.256
Cardiovascular disease, *n* (%)	15 (39.5)	17 (34.7)	13 (32.5)	0.805
Systemic autoimmune disorders, *n* (%)	8 (21.0)	19 (38.7)	0 (0)	<0.001
Type of vaccine				
Pfizer-BioNTech BNT162b2, *n* (%)	34(89.5)	47(95.9)	36 (90)	0.452
Moderna mRNA-1273, *n* (%)	4 (10.5)	2 (4.1)	4 (10)	0.452
Diagnosis				
CHB, *n* (%)	7 (18.4)	23 (46.9)	0 (0)	0.006 ^†^
NAFLD, *n* (%)	9 (23.7)	7 (14.3)	0 (0)	0.262 ^†^
AFLD, *n* (%)	6 (15.8)	0 (0)	0 (0)	0.004 ^†^
AIH, *n* (%)	8 (21.1)	6 (12.2)	0 (0)	0.267 ^†^
PBC, *n* (%)	1 (2.6)	11(22.4)	0 (0)	0.008 ^†^
Hepatic sarcoidosis, *n* (%)	1 (2.6)	0 (0)	0 (0)	0.437 ^†^
CHC, *n* (%)	1 (2.6)	1 (2)	0 (0)	1.000 ^†^
PSC, *n* (%)	3 (7.9)	1 (2)	0 (0)	0.314 ^†^
Budd-Chiari, *n* (%)	1 (2.6)	0 (0)	0 (0)	0.437 ^†^
DILI, *n* (%)	1 (2.6)	0 (0)	0 (0)	0.437 ^†^
Cirrhosis staging scores				
MELD, median (range)	9 (6–25)	NA	NA	NA
CTP, median (range)	6 (5–11)	NA	NA	NA
Immunosuppressive therapy, *n* (%)	12 (31.6)	18 (36.7)	0 (0)	0.616 ^†^
MTX, *n* (%)	1 (2.6)	4 (8.2)	0 (0)	0.381 ^†^
AZA, *n* (%)	6 (15.8)	1 (2)	0 (0)	0.040 ^†^
RTX, *n* (%)	0 (0)	4 (8.2)	0 (0)	0.128 ^†^
MMF, *n* (%)	2 (5.3)	4 (8.2)	0 (0)	0.692 ^†^
TNFi, *n* (%)	1 (2.6)	3 (6.1)	0 (0)	0.629 ^†^
GC, *n* (%)	9 (23.7)	5(10.2)	0 (0)	0.090 ^†^

Abbreviations: PWLD—patients with liver diseases; CHB—chronic hepatitis B infection; NAFLD—non-alcoholic fatty liver disease; AFLD—alcoholic fatty liver disease; AIH—autoimmune hepatitis; PBC—primary biliary cholangitis; MELD—model for end-stage liver disease; CTP—Child–Turcotte–Pugh; MTX—methotrexate; AZA—azathioprine; RTX—rituximab; MMF—mycophenolate mofetil; TNFi—tumor necrosis factor inhibitors; GC—glucocorticoids; NA—not applicable. ^†^ *p* value represents statistical analysis between cirrhotic and non-cirrhotic patients.

## Data Availability

The data presented in this study are available on request from the corresponding author.

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
