# Peer review of "Antibody Responses after SARS-CoV-2 Vaccination in Patients with Liver Diseases"

_viruses, 2022, doi:10.3390/v14020207_

Round 1
Reviewer 1 Report
Strengths:
- The paper provides useful and reassuring data on SARS-CoV-2 mRNA vaccination in individuals with liver disease as a prospective observational study.
Limitations:
- Use of surrogate neutralisation assay - which are known to be limited compared with cell based neutralisation assays
- Antibody data only on wuhan strain and no data on variants of concern
- Lack of T-cell assays
- Small numbers
- From baseline info appears to be mostly fairly moderate rather than severe disease
Specific comments:
Methods:
Line 70
Can you add more details on how side effects were collected. Was this collected retrospectively at visits or were there any symptom diaries? Any solicited symptoms?
Line 72
How were data on breakthrough infections gathered?
Line 76
Need to state which isolate spike protein used in each assay as this is highly relevant given the rise of variants of concern
Line 84
Use of surrogate neutralisation is a limitation rather than cell based neutralisation assays (see https://doi.org/10.1128/JCM.00527-21 )
Lack of t-cell data
Stats - appear appropriate but I am not qualified to comment on multivariable stats
Results
Line 131
"When excluding PLWD under immunosuppressive treatment, no significant differences were found in
seroconversion rates, antibody titers and neutralizing activity levels among all subgroups (data not shown)"
it would be useful to include this data in the manuscript as a supplementary figure
Line 148
If the control volunteers were assessed can they be included here also?
Discussion
Line 209 - is this the assay threshold of detection or another threshold?
Other - would be helpful to mention limitations of surrogate neutralisation assay and lack of variants of concern
Author Response
Reviewer nr 1
Thank you very much for your remarks. Please find the authors’ responses to each one of them.
- Line 70 “Can you add more details on how side effects were collected. Was this collected retrospectively at visits or were there any symptom diaries? Any solicited symptoms?”
Thank you very much for your remark. Side effects were collected retrospectively using a structured questionnaire by each patient at 1 month after 2nd vaccine dose. This is now also included in the revised manuscript.
- How were data on breakthrough infections gathered?
Thank you for your question. Regarding symptomatic breakthrough infections, these were noted at each patient’s follow up visit, up to a 6-months’period after the 2nd vaccine dose. For patients that missed their follow up visit, a telephone interview was performed. This is also added in the manuscript.
- Need to state which isolate spike protein used in each assay as this is highly relevant given the rise of variants of concern
Regarding Anti-SARS-CoV-2 IgG antibody assay, the S1 domain of spike protein of Wuhan-Hu-1 isolate SARS-CoV-2, was used, while for neutralizing antibodies assay, wild-type SARS-CoV-2 RBD HRP was used. Both of these corrections have been also made in the document.
- Use of surrogate neutralisation is a limitation rather than cell based neutralisation assays (see https://doi.org/10.1128/JCM.00527-21 ) and lack of T-cell data
Thank you very much for your remarks. You are perfectly right for both of them. This is now added in study’s limitations in the discussion section.
- Line 131 - "When excluding PLWD under immunosuppressive treatment, no significant differences were found in seroconversion rates, antibody titers and neutralizing activity levels among all subgroups (data not shown)". It would be useful to include this data in the manuscript as a supplementary figure
Thank you for your suggestion. Data has been added as supplementary material
- Line 148 - If the control volunteers were assessed can they be included here also?
Unfortunately healthy volunteers were not re-assesed at 3 months post-vaccination time-point due to the study’s protocol. This is also added in the limitations section.
- Line 209 - is this the assay threshold of detection or another threshold?
Thank you for your question. This is the assay threshold and it is now added in the manuscript.
- Other - would be helpful to mention limitations of surrogate neutralisation assay and lack of variants of concern
Thank you for your remark. These limitations are now added in the manuscript.
Reviewer 2 Report
This study prospectively assessed antibody responses in patients with liver diseases ,with and without cirrhosis ,vaccinated with 2 doses of mRNA
SARS-CoV-2 vaccines (Pfizer-BioNTech 62 BNT162b2 or the Moderna mRNA-1273).Most data for this papulation come from Liver Transplant registries.Although limited by numbers and a binary analysis for the cirrotic population more or less of B8-9 cannot be done,the results are of value to be reported.
Comment : I would like to read thorougly the paper Cirrhosis-associated immune dysfunction:distinctive features and clinical relevance by J Hepatol. 2014 Dec;61(6):1385-96. doi: 10.1016/j.jhep.2014.08.010. Epub 2014 Aug 15.PMID: 25135860
Then ,to introduce to yourmauscript a few comments on the direct or indirect mechanisms of Ab production dysfunction in cirrosis,that may act in the vaccination senario.
Author Response
Reviewer nr 2
I would like to read thorougly the paper Cirrhosis-associated immune dysfunction:distinctive features and clinical relevance by Albillos A, Lario M, Álvarez-Mon M.J Hepatol. 2014 Dec;61(6):1385-96. doi: 10.1016/j.jhep.2014.08.010. Epub 2014 Aug 15.PMID: 25135860
Then ,to introduce to your manuscript a few comments on the direct or indirect mechanisms of Ab production dysfunction in cirrosis,that may act in the vaccination senario.
Thank you very much for your suggestion. We have added a small section in the first paragraph of discussion, talking about possible mechanisms leading to vaccine hypo-responsiveness in cirrhotic patients.